# Reproducibility and Relative Validity of a Short Food Frequency Questionnaire for Chinese Older Adults in Hong Kong

**DOI:** 10.3390/nu16081132

**Published:** 2024-04-11

**Authors:** Vicky Wai-ki Chan, Joson Hao-shen Zhou, Liz Li, Michael Tsz-hin Tse, Jane Jia You, Man-sau Wong, Justina Yat-wa Liu, Kenneth Ka-hei Lo

**Affiliations:** 1Department of Food Science and Nutrition, The Hong Kong Polytechnic University, Hong Kong SAR, China; vickycwk.chan@polyu.edu.hk (V.W.-k.C.); sin-liz.li@polyu.edu.hk (L.L.); man-sau.wong@polyu.edu.hk (M.-s.W.); 2Department of Paediatrics, Faculty of Medicine, The Chinese University of Hong Kong, Hong Kong SAR, China; 1155161736@link.cuhk.edu.hk; 3Jockey Club School of Public Health and Primary Care, The Chinese University of Hong Kong, Hong Kong SAR, China; michael.tszhin.tse@link.cuhk.edu.hk; 4Department of Computing, The Hong Kong Polytechnic University, Hong Kong SAR, China; jane.you@polyu.edu.hk; 5School of Nursing, The Hong Kong Polytechnic University, Hong Kong SAR, China; justina.liu@polyu.edu.hk; 6Research Institute for Smart Ageing, The Hong Kong Polytechnic University, Hong Kong SAR, China

**Keywords:** dietary assessment, food frequency questionnaire, Hong Kong, older adults, relative validity, reproducibility

## Abstract

Changes in an individual’s digestive system, hormones, senses of smell and taste, and energy requirement accompanying aging could lead to impaired appetite, but older adults may not notice their risk of nutrient deficiency. When assessing the dietary intake of older adults, it was found that they had more difficulties with short-term recall and open-ended recall and would experience greater fatigue and frustration when compared to younger individuals when completing a lengthy questionnaire. There is a need to develop a brief dietary assessment tool to examine the nutritional needs of older adults. In this study, we aimed to assess the diet of Hong Kong older adults using the short FFQ and examine its reproducibility and relative validity as a dietary assessment tool. Dietary data of 198 older adults were collected via FFQs and three-day dietary records. Correlation analyses, cross-tabulation, one-sample *t*-tests, and linear regression analyses were used to evaluate the relative validity of the short FFQ. In general, the short FFQ was accurate in assessing the intake of phosphorus, water, grains, and wine, as shown by a significant correlation (>0.7) between values reported in the FFQs and dietary records; good agreement (more than 50% of observations belonged to the same quartile) and insignificant differences detected with the one-sample *t*-tests and linear regression analyses were observed for the above four variables. Additionally, the intake of proteins, carbohydrates, total fat, magnesium, and eggs in terms of the values reported in the FFQs and dietary records showed good agreement.

## 1. Introduction

The World Health Organization (WHO) has defined healthy aging as ‘the process of developing and maintaining the functional ability that enables wellbeing in older age’ and has recognized the importance of adequate nutrition in preventing and managing non-communicable diseases [1]. Changes in the digestive system, hormones, senses of smell and taste, and energy requirements accompanying aging could lead to impaired appetite, but older adults may not notice their risk of nutrient deficiency [2]. With reference to a previous cross-sectional study with 179 Hong Kong older adults aged 60 or above, around 30% of them were at risk of malnutrition [3]. Another meta-analysis with 19,938 Chinese community-dwelling older adults aged 60 or above revealed that about 41.2% of them were either malnourished or at risk of malnutrition [4].

Malnutrition, as defined by the WHO, refers to deficient, excessive, or imbalanced energy and/or nutrient intakes in an individual [5]. It is associated with numerous non-communicable diseases (NCDs), such as hypertension, cardiometabolic disorders, type II diabetes mellitus (T2DM), dementia, and even cancers [6,7,8]. An imbalance in nutrition could potentially trigger insulin resistance, oxidative stress, and inflammation, which ultimately contribute to NCD development [9]. Due to its aging population, Hong Kong is facing a mounting challenge of NCDs. The proportions of registered deaths due to cancers and heart diseases surged from 12.2% and 9.4% in 1961 to 30.5% and 13.3% in 2016 [10]. Notably, in 2016, around 55% registered deaths in Hong Kong were due to NCDs, including cardiovascular diseases, cancers, diabetes, and chronic respiratory diseases [10]. As projected by the Health Bureau of the Hong Kong government, the total health expenditure in relation to gross domestic production (GDP) will increase from 5.3% (HKD 67.8 billion) in 2004 to 9.2% (HKD 315.2 billion) in 2033 [11]. When the disease burden increases with the aging population in Hong Kong, it is especially important for older adults to maintain healthy diets.

To assess dietary intake at the population level, three-day dietary records and Food Frequency Questionnaires (FFQs) are commonly used tools. While a dietary record can assess the foods and beverages being consumed, it puts a burden on respondents due to the high recall required, especially for older adults, who have more difficulties with short-term recall and open-ended recall [12]. Moreover, an FFQ evaluates usual dietary intake, but it can be quite lengthy to administer (containing 100–200 items and taking around an hour for completion) and requires integrated information from multiple occasions of food consumption. Older adults were found to have longer response times and potentially experience greater fatigue and frustration when compared to younger individuals when completing lengthy questionnaires or interviews [12]. This indicates the need to develop a user-friendly tool to assess the dietary intakes of older adults.

In Hong Kong, a short version of the FFQ was validated for 103 participants from disadvantaged communities, which showed good validity in reflecting nutrient intakes, including water and total energy [13]. Nevertheless, the previous validation study was nested within a dietary intervention study [14], and therefore the reproducibility of the short FFQ was not evaluated [13]. Furthermore, the previous 50-item FFQ did not differentiate between high-fat and low-fat meat categories, leading to data discrepancies in accurately assessing the intake of energy from fat [13]. Therefore, the current FFQ has been revised to better capture the dietary habits of older adults, which might be different from the disadvantaged communities previously studied. The aim of this study is to assess the diets of Hong Kong older adults using the short FFQ and examine its reproducibility and relative validity as a dietary assessment tool for the general older population.

## 2. Materials and Methods

### 2.1. Study Design and Participants

In collaboration with the Research Centre for Gerontology and Family Studies (RCGFS) and/or district community health centers, 200 older adults were included in this cross-sectional study during the COVID-19 pandemic. According to previous research, the proposed sample size for the present study was sufficient to detect the correlation between dietary intakes assessed with the FFQ and dietary records for validation purposes [15]. The design of the present validation study was planned with reference to several similar studies [16,17,18]. Participants were considered eligible if they fulfilled the following criteria: (i) Hong Kong Chinese citizen; (ii) permanent resident in Hong Kong; (iii) aged 50–79 (this range referred to previous studies conducted among older adults in China [19,20,21]); (iv) able to speak and understand Chinese; (v) willing to follow the study procedures. Participants who (i) were currently participating in any clinical trial or trial with dietary intervention; (ii) required a specialized diet (e.g., kidney disease, diabetes mellitus); or (iii) had any major medical or psychological condition, as judged by the investigators, were excluded from this study.

### 2.2. Data Collection

People who met the sample selection criteria and were interested in participating were approached to obtain their informed consent through a face-to-face interview or electronic methods [22,23] (e.g., smartphone, fax, or email [24]). The participants’ information was self-reported using questionnaires. Demographic characteristics, including sex, age, education level, total family income, smoking status, and medical history, were collected using a standardized questionnaire.

### 2.3. Dietary Assessments

Dietary intakes were assessed using two dietary assessment methods, including three 24 h dietary records and a newly developed FFQ. Practical instructions to complete the FFQ and dietary records were provided. The 60-item FFQ (<30 min for administration) with food items suitable for Hong Kong older adults was newly developed based on clinical experience from a panel of dietitians, nutritionists, and nurses, and with reference to an FFQ available for the local adult population [25]. Foods with similar nutrient contents were combined to form a shortened FFQ [26] that attempted to cover the full picture of the participants’ dietary intake [27]. For example, rice, noodles, macaroni, pasta, and rice noodles were combined into the refined grains category. Considering the nutritional differences between some similar foods, e.g., full-fat versions and low-fat versions, the foods’ key characteristics and specific nutrients of concern were the major grouping criteria [15,28]. Related food groups were listed adjacently to enhance the respondents’ recall of their food consumption [26].

In the form of a face-to-face or phone interview, each participant was asked to complete the FFQ twice within 3 to 4 weeks (for reproducibility evaluation), and three 24 h dietary records in between the two administrations of the short FFQ as a reference method to validate the FFQ (for the validity evaluation). The three 24 h dietary records included two weekdays and one weekend day.

The food group intake frequency was reported per day, week, or month, and portions were reported based on standard portion sizes, as pieces, glasses, cups, spoons, milliliters, or grams. On the same interview day, each participant was also asked by trained research staff to recall all foods and beverages that they had consumed over the past 24 h. Food photo albums and eating utensils demonstrating standardized portions were displayed to aid recall. The 2nd administration of the FFQ (FFQ2) was carried out 3 to 4 weeks after the 1st administration of the FFQ. The FFQ2 and the food portion booklet as well as the additional 24 h recall forms were given to the participants via email, smartphone messages, or post within 3 to 4 weeks after the 1st administration of the FFQ (FFQ1). Daily dietary and nutrient intake collected by both the FFQs and 24 h records were entered and calculated using the nutrition analysis software Food Processor Nutrition Analysis software version 11.11 (ESHA Research, Salem, MA, USA), including local foods selected from food composition tables from China and Hong Kong.

### 2.4. Statistical Analyses

Only those who had completed two FFQs and three-day dietary records were included for analysis. Continuous variables were presented as the mean (standard deviations) for parametric data and as the median (interquartile range) for nonparametric data. Categorical variables were presented as a number (percentage). Intraclass correlation coefficients (ICCs) were calculated to examine the reproducibility between FFQ1 and FFQ2. Based on the reported ICCs, the reliability of the dietary intake data could be categorized into poor (<0.5), moderate (0.5 to 0.75), good (0.75 to 0.90), and excellent (>0.90), respectively [29].

Pearson’s correlation was used to validate the results of FFQ1 against the average of three 24 h records. As an alternative analysis, we applied energy adjustment using the residual method of Willett [30] and computed the Pearson correlation using the energy-adjusted nutrient and food group intakes. Differences in nutrient and food group intakes between the first FFQ and the average of three 24 h records were examined using one-sample *t*-tests. We also examined if the differences in nutrient and food group intakes between the first FFQ and the average of three 24 h records increased with the average of the FFQ1 results and 24 h records, with the use of linear regression. A quartile classification analysis was used to categorize the nutrient intakes and food group intakes calculated from the FFQ and the three 24 h records. The following is the list of categories for the nutrient intake: total energy, energy from total fat, energy from saturated fat, energy from trans fat, protein, carbohydrates, total dietary fiber, total sugar, cholesterol, water, vitamin C, calcium, copper, iron, magnesium, manganese, phosphorus, potassium, sodium, and zinc.

In addition, the percentages of subjects correctly classified into the same, adjacent (±1 quartile), or extreme quintiles (±3 quartiles) were calculated. Lastly, for the total energy intake and three common macronutrients, including carbohydrates, protein, and total fat, Bland–Altman plots were performed to visually present the agreement between the first FFQ and the three 24 h records. All statistical analyses were performed using the statistical package SPSS version 24.0 (SPSS Inc., Chicago, IL, USA). All statistical tests were two-tailed, and significance was set at *p* < 0.05.

## 3. Results

### 3.1. Demographic Characteristics

A total of 198 participants (66.7% female) completed the two short FFQs and three-day dietary records. Demographic characteristics of all participants are summarized in Table 1. Their average age and BMI were 64.0 ± 6.0 years old and 22.8 ± 3.4 kg/m^2^, respectively. Amongst these 198 participants, 63.6% of them were married, 85.4% had received senior secondary education or above, 90.9% were never-smokers, 41.4% exercised more than five times per week, 52% took supplements frequently, 46% were healthy without any diseases, and 36.4% had a total family income under HKD 10,000. On average, all participants completed the two FFQs 25 days apart and the average completion time required for the second FFQ (27.3 ± 7.2 min) was shorter than that for the first FFQ (33.9 ± 7.5 min).

### 3.2. Dietary Intakes of the Included Participants

The prevalence of excessive or deficient intakes of 20 nutrients among the 198 participants was determined using dietary data obtained from their three-day dietary records. These results are shown in Table 2. Overall, a majority of participants had an excessive intake of sodium (93.2%), energy from total fat (74.8%), and cholesterol (68.9%). Conversely, they had a notably deficient intake of total energy (68.9%), total dietary fiber (84.5%), water (64.1%), calcium (74.8%), and manganese (76.7%).

### 3.3. Reproducibility of the Short FFQ and Its Relative Validity with Three-Day Dietary Records

#### 3.3.1. Reproducibility of the Short FFQ

The reproducibility of the short FFQ, administered twice, 25 days apart, for assessing nutrient and food group intakes was evaluated and these results are shown in Table 3 and Table 4. For energy and nutrient intakes (Table 3), the ICCs ranged from 0.85 to 0.96. Most of the parameters demonstrated excellent reliability (>0.90), except for the intakes of energy from trans fat (0.90), trans fat (0.90), and sodium (0.85), which demonstrated good reliability. The results were still similar after energy adjustment. For the food group intake (Table 4), the ICCs ranged from 0.76 (legumes and savory snacks) to 0.98 (wine). Out of the 20 food groups, 11 food groups demonstrated excellent reliability (>0.90), while the remaining 9 food groups demonstrated good reliability (0.75 to 0.90). After energy adjustment, the reliability of four food groups (i.e., grains; meat, poultry, processed meat, and organ meat; nuts and seeds; and oil) improved from good to excellent, while that of two food groups (i.e., sugary snacks and dairy/dairy products) changed from excellent to good.

#### 3.3.2. Analyzing the Consistency of Dietary Data Reported by FFQs and Three-Day Dietary Records Using Bland–Altman Plots

The differences in the measurements of total energy (Figure 1), protein (Figure 2), carbohydrates (Figure 3), and total fat (Figure 4) between the FFQ1 and three-day dietary records were visualized using Bland–Altman plots. A total of nine outliers can be observed in both Figure 1 and Figure 3. In Figure 2, 13 outliers are identified, and 11 outliers are depicted in Figure 4. The Bland–Altman analysis results for the remaining nutrient parameters are summarized in Appendix A. The percentage of outliers for the nutrients ranged from 3.03% (zinc) to 7.07% (calcium). To better illustrate the limit of agreement between the short FFQ and the three-day dietary records, Bland–Altman plots are provided in Appendix A.

#### 3.3.3. Correlations between the Short FFQ1 and Three-Day Dietary Records

The correlations between the FFQ and three-day dietary records regarding the nutrient and food group intakes are shown in Table 5 and Table 6, respectively. For the energy and nutrient intakes (Table 5), significant positive correlations (r ≥ 0.7) were found in 12 parameters, namely total energy, energy from total fat, energy from saturated fat, protein, carbohydrates, total dietary fiber, total fat, saturated fat, water, magnesium, phosphorus, and potassium. After energy adjustment, all parameters maintained a consistent correlation. For the food group intakes (Table 6), significant positive correlations (r ≥ 0.7) were found in seven parameters, namely grains, fruits, vegetables, eggs, nuts and seeds, water, and wine. These results remained similar after energy adjustment.

#### 3.3.4. Cross-Classification of Dietary Intakes between FFQ1 and Three-Day Dietary Records

The dietary data from FFQ1 and the three-day dietary records were cross-tabulated to examine the agreement between dietary intakes at the nutrient and food group levels. These results are displayed in Table 7 and Table 8, respectively. For the crude mean energy and nutrient intakes, the percentage of agreement within the same quartile varied from 39.4% (vitamin C) to 62.6% (carbohydrates). In general, around 61% of the crude intakes estimated from the FFQs and three-day dietary records were within the same or adjacent quartiles. After energy adjustment, substantial changes in the percentage of agreement within the same quartile were observed in the following two variables: carbohydrates (from 62.6% to 37.4%) and trans fat (from 44.4% to 34.8%). In general, the percentage of agreement within the same or adjacent quartiles was 55% after adjusting for energy. For the crude mean food group intakes, the percentage of agreement within the same quartile varied from 27.8% (condiments and savory snacks) to 74.7% (wine). In general, around 73% of the crude intakes estimated from the FFQs and three-day dietary records were within the same or adjacent quartiles. After energy adjustment, substantial changes in the percentage of agreement within the same quartile were observed in the following two variables: eggs (from 59.6% to 40.9%) and savory snacks (from 61.1% to 27.8%). In general, the percentage of agreement within the same or adjacent quartiles was 81% after adjusting for energy.

#### 3.3.5. One-Sample *t*-Tests and Linear Regression Analyses for FFQ1 and Three-Day Dietary Records

In Table 9, the significance of differences between the dietary intakes reported in the FFQ1 and three-day dietary records at the nutrient and food group levels was determined using one-sample *t*-tests, respectively, and a *p*-value ≤ 0.05 indicated a significant difference. For the energy and nutrient intakes, insignificant differences were found in the following six parameters: total sugar, water, vitamin C, magnesium, phosphorus, and zinc. For the food group intakes, insignificant differences were found in the following seven parameters: grains, eggs, fish and seafoods, tea or coffee, water, savory snacks, and wine. Furthermore, linear regression analyses were conducted to evaluate the potential significance (*p*-value ≤ 0.05) of the differences between the dietary intakes obtained from FFQ1 and the three-day dietary records in relation to the levels of dietary intake. For the energy and nutrient intakes, insignificant differences were found in the following seven parameters: protein, carbohydrates, total fat, iron, water, manganese, and phosphorus. For the food group intakes, insignificant differences were found in the following six parameters: grains, beverages, soy and soy products, legumes, water, and wine.

## 4. Discussion

### 4.1. Summary of Main Findings

In the present study, the reproducibility and validity of a short version of the FFQ for assessing the dietary intakes of 198 Chinese older adults in Hong Kong, as well as the prevalence of excessive or deficient nutrient intakes among them, were evaluated. In general, most participants had an excessive intake of sodium, energy from total fat, and cholesterol, while they had a notably deficient intake of total energy, total dietary fiber, water, calcium, manganese, and magnesium. From the ICC results, the reliability of both the nutrient and food group intake data was good to excellent. The relative validity of the short FFQ at the nutrient and food group levels was revealed through correlation analyses, cross-classification of the dietary data collected from the FFQs and three-day dietary records, one-sample *t*-tests, and linear regression analyses, respectively. For the nutrient intakes, water and phosphorus demonstrated good agreement in all analyses, while protein, carbohydrates, total fat, and magnesium showed good agreement in most analyses. For the food group intakes, grains, water, and wine demonstrated good agreement in all analyses, while eggs showed good agreement in most analyses.

### 4.2. Dietary Intakes of the Included Participants

The prevalence of dietary intakes violating the Chinese DRI or WHO recommendations was calculated based on data collected from three-day dietary records. Overall, most participants reported excessive intakes of energy from total fat (74.8%), cholesterol (68.9%), and sodium (93.2%). This implied that they had a high-fat and high-sodium diet. Their average sodium intake was 3032.4 mg per day, which exceeds the WHO-recommended sodium intake (2000 mg per day) [31] and is comparable to the mean sodium intake (3520 mg per day) of adults in Hong Kong [32]. The Department of Health has indicated that the frequent practice of eating out and the consumption of preserved food were potential contributors to the elevated sodium intake observed among the Hong Kong population [32]. Iodized salt is readily available in numerous countries, including China, the United States, and New Zealand, but not in Hong Kong [33]. To reduce sodium intake and combat iodine deficiency disorders, the WHO has recommended the implementation of universal salt iodization [34]. According to the Population Health Survey of 2020–2022 conducted by the Department of Health in Hong Kong [35], both males and females aged 15 to 84 had mean urinary iodine concentrations <100 μg/L, indicating an insufficient iodine intake and a mild iodine deficiency. This highlights the importance of salt iodization as a measure to reduce sodium intake and prevent iodine deficiency.

In contrast, more than 80% of the participants did not report an adequate intake of total dietary fiber. Chinese older adults are advised to consume 25 g of dietary fiber per day [36]. However, the average dietary fiber intake of the participants in this study was only 17.1 g per day. Dietary fiber has been found to have an inverse association with all-cause and cardiovascular mortality in older adults as it can promote the growth of healthy gut microbiota, which in turn provides metabolic benefits such as improved glucose and lipid metabolism, enhanced insulin sensitivity, and reduced abdominal adiposity [37]. It can also stimulate the production of several gut hormones, adipokines, and bile acids, which in turn can improve metabolic health [37]. Future studies may focus on the barriers preventing adequate dietary fiber consumption among Hong Kong older adults, which could help develop health promotion strategies.

In addition, around 75% of participants failed to consume enough calcium. For Chinese adults aged 50 or above, the recommended calcium intake is 800 milligrams per day, while the average calcium intake of the participants in this study was 636.1 milligrams per day. In the first Total Diet Study performed by the Centre for Food Safety in Hong Kong, more than 90% of adults failed to meet the recommended calcium intake [38]. As suggested, a comparatively lower consumption of dairy products among the Hong Kong population was one of the potential contributing factors [38]. Calcium is important for preserving bone mass and preventing osteoporosis [39]. Considering the reduced rate of calcium absorption during later stages of life, it is essential for older adults to ensure an adequate intake of calcium in order to prevent bone loss [39].

In addition to dietary fiber and calcium intake, most participants also reported insufficient intakes of total energy, water, magnesium, and manganese. With aging, numerous physical functions tend to decline to varying extents. The decline in digestion and renal functions among older adults can affect their water absorption from food [40]. Additionally, a blunted sensation of thirst can also lead to reduced water intake [40]. With the gradual loss of muscle mass during aging, the energy demands, metabolic rates, and physical activity levels of older adults decrease, which may lower their food and energy intakes [40]. Since dietary fiber-rich food tends to be a good source of magnesium [41], an insufficient intake of dietary fiber may elevate the risk of magnesium deficiency.

### 4.3. Reproducibility and Relative Validity of the Short FFQ

The reproducibility of the short FFQ was indicated by ICCs, which were categorized into poor (<0.5), moderate (0.5 to 0.75), good (0.75 to 0.90), and excellent (>0.90), respectively [29]. Across the nutrients and food groups, it demonstrated good to excellent reliability.

Pearson correlation coefficients ranging from 0.5 to 0.7 indicate a good correlation, while those larger than 0.7 indicate a very strong correlation between the two methods [42]. At the nutrient level, for most parameters, the correlations between the two methods ranged from being good to very strong. Except for the intakes of energy from trans fat, trans fat, zinc, and sodium, only acceptable correlations were observed. At the food group level, the two methods demonstrated good to very strong correlations for most parameters, except for condiments, legumes, sugary snacks, savory snacks, and dim sum. The agreement between the two dietary assessment methods was also assessed using cross-classification. Most of the reported values for various parameters fell within the same or adjacent quartiles across the nutrients and food groups, indicating a high level of agreement between the two methods. Furthermore, one-sample *t*-tests and linear regression analyses at both the nutrient and food group levels were performed to detect insignificant differences between the two assessment methods. Insignificant differences were detected in six nutrient parameters, namely total sugar, water, vitamin C, magnesium, phosphorus, and zinc, and in seven food group parameters, namely grains, fish and seafoods, eggs, tea or coffee, water, savory snacks, and wine, which further supported the agreement in the reported values.

After applying various statistical validation methods, it was found that the short FFQ was particularly effective in accurately assessing the participants’ intakes of phosphorus, water, grains, and wine, which showed a significant correlation (>0.7) between the values obtained with the two assessment methods; good agreement (more than 50% of observations belonging to the same quartile) and insignificant differences detected with the one-sample *t*-tests and linear regression analyses were observed for the above four variables. Additionally, the short FFQ showed accuracy in capturing the intakes of proteins (with a significant correlation of 0.76, good agreement, and insignificant differences between the assessment methods according to the linear regression analysis), carbohydrates (with a significant correlation of 0.79, good agreement, and insignificant differences between the assessment methods according to the linear regression analysis), total fat (with a significant correlation of 0.75, good agreement, and insignificant differences between the assessment methods according to the linear regression analysis), magnesium (with a significant correlation of 0.76, good agreement, and insignificant differences between the assessment methods according to the one-sample *t*-test), and eggs (with a significant correlation of 0.74, good agreement, and insignificant differences between the assessment methods according to the one-sample *t*-test).

### 4.4. Comparison between the Previous and Current Short FFQs Used in the Hong Kong Community

When compared to the previous short FFQ used for disadvantaged communities [13], the current FFQ has been revised to capture a broader range of dietary factors and provide a more comprehensive and accurate assessment of dietary intakes. Specific changes include (1) the addition of a category for soft or thin refined grains; (2) the separation of leafy vegetables, melons, and root vegetables to investigate the effects of nutrients and differentiate starchy vegetables; (3) a classification between fatty meat and lean meat; (4) differentiation between poultry with or without skin; (5) a classification between oily fish and other fish; (6) the replacement of animal liver with animal organs to encompass a wider range of animal organ consumption; (7) differentiation between high- or low-fat tofu and soy products; (8) the separation of mushrooms and algae; (9) the replacement of sweetened cookies with all types of biscuits; (10) the removal of fried dishes to avoid overlap with other items; and (11) the separation of low-sugar or diet drinks. Additional categories have also been added, including egg whites, low-sugar tea or coffee, alcoholic drinks, and added sugars.

The fish category was separated into oily fish or other fish due to the differences in their vitamin D and omega-3 contents. Older adults are at higher risk of vitamin D deficiency as its dietary sources are very limited (i.e., fortified milk and oily fish), while they have a declined ability to synthesize vitamin D through their skin after sunlight exposure [43]. It was suggested that low vitamin D levels are negatively associated with cognitive function and vitamin D consumption could be a potential therapy to fight against cognitive decline and dementia during aging [44]. Similar to vitamin D, the food sources of omega-3 fatty acids in a standard diet are limited (i.e., oily fish, flax seeds, and walnuts), while omega-3 fatty acids obtained from flax seeds and walnuts might not be as beneficial as those obtained from oily fish [43]. With reference to a previous review, omega-3 fatty acids may be beneficial for maintaining cognitive function, bone health, immune function, and muscle performance during aging [45].

Vitamin B6 is another dietary factor of concern among older people as they have a reduced food intake and vitamin B6 absorption rate, while having increased catabolism and impaired phosphorylation [46]. An inadequate vitamin B6 intake could impair immune or even cognitive function [43]. Therefore, the animal liver category was replaced by animal organs as they are also good sources of vitamin B6, and the vegetables category was divided into leafy vegetables, melons, and root vegetables to better capture the intake of vitamin B6 among older adults. This separation of the vegetables category also allowed us to better capture the folate intake, as its content in green leafy vegetables is much higher than in other vegetables. An adequate folate intake is essential as too much of it can mask vitamin B12 deficiency, while not having enough can increase the risk of cognitive decline during aging [43]. Alcoholic drink consumption was added to the FFQ as it can alter the absorption of micronutrients such as iron, folate, thiamine, and zinc [47] and is associated with a higher risk of osteoporotic fractures [48]. Meat, poultry, tofu, and soy products were categorized into low- or high-fat categories, respectively. In consideration of their altered body compositions due to aging, older adults are at higher risk of sarcopenic obesity, which can contribute to several metabolic diseases, such as cardiovascular diseases, diabetes, and cancers [49]. Therefore, the categorization between low-fat and high-fat foods allowed a more accurate evaluation of their energy balance between intake and expenditure. The intake of added sugars was found to be positively associated with an increasing risk of frailty and worsening cardiometabolic risk factors among older adults [50]. Hence, to better capture the sugar intakes, low-sugar and diet drinks were separated into two categories, while two new categories, namely low-sugar tea or coffee and extra sugar, were added to the FFQ.

### 4.5. Limitations

A few limitations of the present study should be noted, although the short FFQ has shown good agreement in assessing several nutrient and food group intakes of Chinese older adults. First of all, since this study was performed during the COVID-19 pandemic, the first FFQ was completed through phone calls, and participants were provided with photos of standardized eating utensils as references for reporting portion sizes. The second FFQ was completed either face-to-face or through phone calls. In each face-to-face session, standardized eating utensils, instead of photos, and food models were provided to help the participants estimate their consumption portions. Therefore, the chance of misreporting may be higher for those who chose phone calls. Second, since the participants were allowed to choose three days and record their dietary intakes themselves, the dietary records may not be representative in capturing their typical diets. Third, since most of the participants were female, the results may not represent the general Chinese older population in Hong Kong. Nevertheless, our results have revealed the potential of this short version of the FFQ for assessing nutritional needs at the nutrient and food group levels among Hong Kong older adults.

## 5. Conclusions

In this validation study, the present short FFQ has demonstrated its potential for assessing the dietary intakes of Chinese older adults, especially for phosphorus, water, grains, and wine. The FFQ also showed good agreement in capturing their intakes of proteins, carbohydrates, total fat, magnesium, and eggs.

## Figures and Tables

**Figure 1 nutrients-16-01132-f001:**
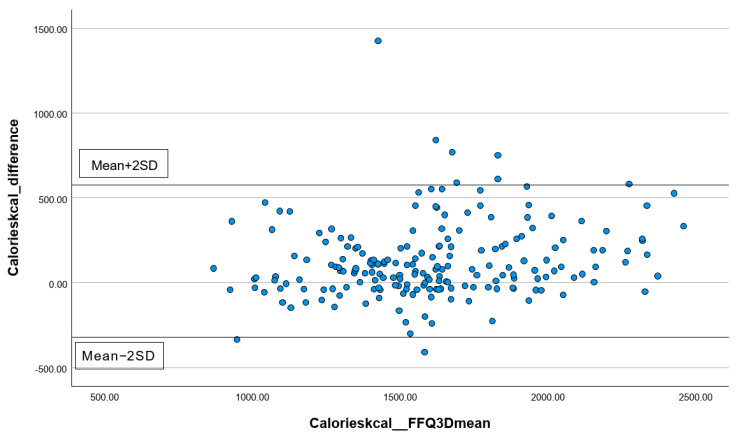
Bland–Altman plot of the total energy data. Abbreviations: FFQ: Food Frequency Questionnaire; SD: standard deviation; 3D: three-day dietary records.

**Figure 2 nutrients-16-01132-f002:**
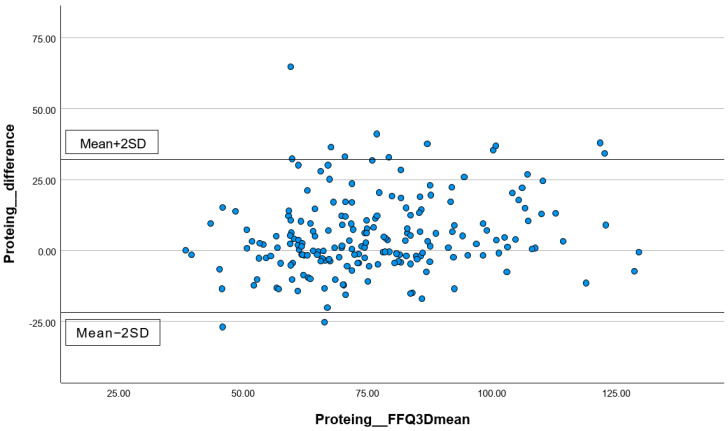
Bland–Altman plot of the protein data. Abbreviations: FFQ: Food Frequency Questionnaire; SD: standard deviation; 3D: three-day dietary records.

**Figure 3 nutrients-16-01132-f003:**
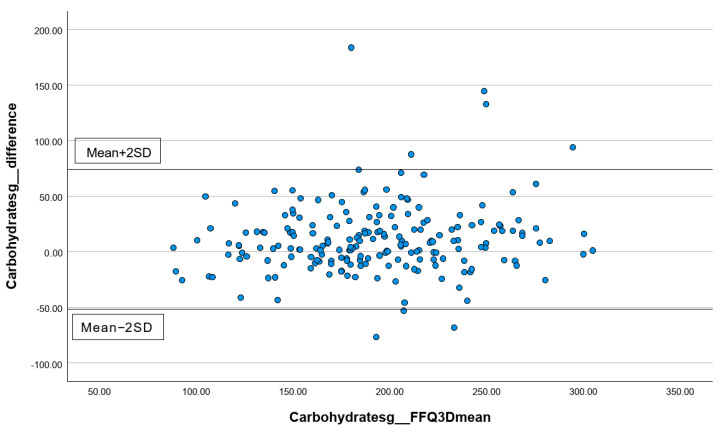
Bland–Altman plot of the data on carbohydrates. Abbreviations: FFQ: Food Frequency Questionnaire; SD: standard deviation; 3D: three-day dietary records.

**Figure 4 nutrients-16-01132-f004:**
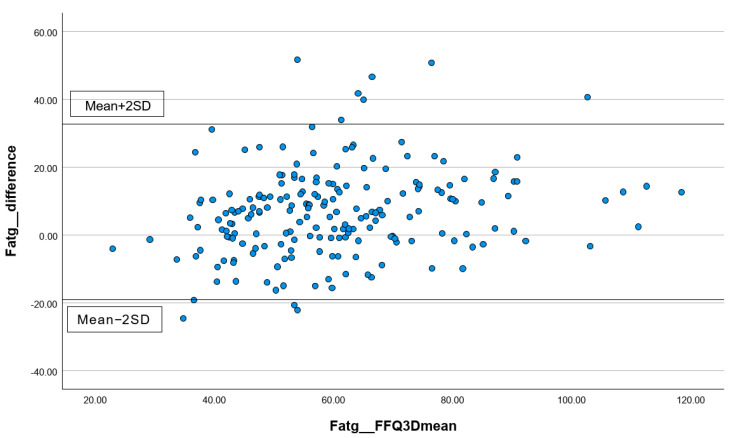
Bland–Altman plot of the total fat data. Abbreviations: FFQ: Food Frequency Questionnaire; SD: standard deviation; 3D: three-day dietary records.

**Table 1 nutrients-16-01132-t001:** Characteristics of the 198 participants.

Characteristics	Mean (±SD)/N (%)		
	Total (*n* = 198)	Male (*n* = 66)	Female (*n* = 132)
Age (years)	64.0 ± 6.0	66.0 ± 5.3	63.5 ± 5.7
BMI at enrolment (kg/m^2^)	22.8 ± 3.4	24.0 ± 3.1	22.0 ± 3.5
Education			
Junior Secondary or below	29 (14.6%)	5 (7.5%)	24 (18.2%)
Senior Secondary	62 (31.3%)	17 (25.8%)	45 (34.1%)
Post-Secondary	36 (18.2%)	13 (19.7%)	23 (17.4%)
Tertiary or above	71 (35.9%)	31 (47.0%)	40 (30.3%)
Marital status			
Not married ^a^	72 (36.4%)	13 (19.7%)	59 (44.7%)
Married	126 (63.6%)	53 (80.3%)	73 (55.3%)
Total family income (HKD)			
<10,000	72 (36.4%)	20 (30.3%)	52 (39.4%)
10,000–19,999	26 (13.1%)	9 (13.6%)	17 (12.9%)
20,000–29,999	23 (11.6%)	6 (9.1%)	17 (12.9%)
30,000–39,999	24 (12.1%)	9 (13.6%)	15 (11.4%)
>40,000	53 (26.8%)	22 (33.3%)	31 (23.5%)
Smoking status			
Smoker	5 (2.5%)	3 (4.5%)	2 (1.5%)
Former smoker	13 (6.6%)	11 (16.7%)	2 (1.5%)
Never-smoker	180 (90.9%)	52 (78.8%)	128 (97.0%)
Exercise habits			
<1 time per week	23 (11.6%)	10 (15.2%)	13 (9.8%)
1–2 times per week	37 (18.7%)	12 (18.2%)	25 (18.9%)
3–4 times per week	56 (28.3%)	18 (27.3%)	38 (28.8%)
>5 times per week	82 (41.4%)	26 (39.4%)	56 (42.4%)
Supplement consumption frequency			
<1 time per week	76 (38.4%)	32 (48.5%)	44 (33.3%)
1–2 times per week	6 (3.0%)	3 (4.5%)	3 (2.3%)
3–4 times per week	13 (6.6%)	2 (3.0%)	11 (8.3%)
>5 times per week	103 (52.0%)	29 (43.9%)	74 (56.1%)
Medical history ^b^			
None	91 (46.0%)	24 (36.4%)	67 (50.8%)
Diabetes	19 (9.6%)	13 (19.7%)	6 (4.5%)
Hypertension	45 (22.7%)	19 (28.8%)	26 (19.7%)
Hyperlipidemia	36 (18.2%	14 (21.2%)	22 (16.7%)
Arteriosclerosis	2 (1.0%)	2 (3.0%)	0 (0%)
Stroke	2 (1.0%)	1 (1.5%)	1 (0.8%)
Cancers	8 (4.0%)	2 (3.0%)	6 (4.5%)
Others	13 (6.6%)	1 (1.5%)	12 (9.1%)
FFQ completion time (minutes)			
1st FFQ	33.9 ± 7.5	34.6 ± 7.7	33.6 ± 7.4
2nd FFQ	27.3 ± 7.2	28.2 ± 8.1	26.9 ± 6.7
Intervals between the two FFQs (days)	25.1 ± 3.6	25.3 ± 3.8	25.0 ± 3.5

^a^ Widowed, divorced, and separated marriages were also included. ^b^ These values might not necessarily add up to 100% as a participant could have more than one disease. Abbreviations: body mass index (BMI); Food Frequency Questionnaire (FFQ); standard deviation (SD).

**Table 2 nutrients-16-01132-t002:** Prevalence of excessive or deficient dietary intakes.

Parameters	Prevalence (%)
Above DRIs	
Energy from Total Fat ^2^	74.8
Energy from Saturated Fat ^2^	29.1
Energy from Trans Fat ^4^	2.91
Total Sugar ^2^	58.3
Cholesterol ^4^	68.9
Sodium ^4^	93.2
Below DRIs	
Total Energy ^1^	68.9
Protein ^5^	6.8
Carbohydrates ^5^	3.9
Total Dietary Fiber ^3^	84.5
Water ^3^	64.1
Vitamin C ^5^	32.0
Calcium ^5^	74.8
Copper ^5^	0.0
Iron ^5^	35.0
Magnesium ^5^	52.4
Manganese ^3^	76.7
Phosphorus ^5^	1.0
Potassium ^3^	23.3
Zinc ^5^	33.0

^1^ Prevalence based on Estimated Energy Requirement (EER). ^2^ Prevalence based on Acceptable Macronutrient Distribution Range (AMDR). ^3^ Prevalence based on adequate intake (AI). ^4^ Prevalence based on the World Health Organization (WHO)-recommended daily intake. ^5^ Prevalence based on Estimated Average Requirement (EAR). Abbreviation: DRIs, dietary reference intakes.

**Table 3 nutrients-16-01132-t003:** Mean daily intakes of energy and selected nutrients and intraclass correlation coefficients between the two FFQs administered to the study participants (*n* = 198).

Parameters	FFQ1					FFQ2					ICCs	
	Mean	SD	Median	25th	75th	Mean	SD	Median	25th	75th	Crude	Energy-Adjusted
Energy (kcal)	1541.2	334.0	1524.4	1321.6	1753.7	1536.5	331.8	1503.9	1315.3	1745.4	0.96 **	--
Energy from Total Fat (kcal)	509.6	147.9	488.0	397.0	589.9	507.4	151.4	482.7	388.8	601.7	0.95 **	--
Energy from Saturated Fat (kcal)	130.4	40.7	127.1	100.6	148.9	128.1	39.5	121.4	99.5	149.7	0.96 **	--
Energy from Trans Fat (kcal)	3.5	2.1	3.0	2.1	4.3	3.5	2.1	2.9	2.1	4.2	0.90 **	--
Protein (g)	73.8	17.9	72.2	61.0	83.7	72.7	17.7	70.5	60.8	83.1	0.94 **	0.93 **
Carbohydrates (g)	186.6	46.6	183.0	152.7	220.7	187.0	44.8	185.8	152.6	214.5	0.95 **	0.93 **
Total Dietary Fiber (g)	15.9	5.2	15.4	12.6	19.1	15.9	5.2	15.2	12.3	18.8	0.96 **	0.97 **
Total Sugar (g)	51.9	18.2	50.1	37.6	62.7	52.0	17.9	50.3	39.3	62.1	0.94 **	0.94 **
Total Fat (g)	56.9	16.6	54.3	44.3	65.8	56.7	17.0	53.8	43.5	67.3	0.95 **	0.93 **
Saturated Fat (g)	14.5	4.5	14.1	11.1	16.5	14.2	4.4	13.5	11.1	16.6	0.96 **	0.95 **
Trans Fat (g)	0.4	0.2	0.3	0.2	0.4	0.4	0.2	0.3	0.2	0.5	0.90 **	0.87 **
Cholesterol (mg)	290.2	93.1	286.1	218.6	353.1	287.4	92.9	278.7	219.5	349.3	0.94 **	0.94 **
Water (g)	2537.7	593.6	2488.9	2094.7	2894.9	2538.8	598.3	2462.8	2145.2	2878.6	0.93 **	0.92 **
Vitamin C (mg)	119.1	43.9	114.5	89.0	144.7	120.2	43.6	115.1	90.8	141.8	0.95 **	0.95 **
Calcium (mg)	685.9	224.9	631.5	547.0	803.5	653.7	221.4	653.7	537.5	774.6	0.94 **	0.95 **
Copper (mg)	1.0	0.3	0.9	0.7	1.1	0.9	0.3	0.9	0.7	1.1	0.93 **	0.94 **
Iron (mg)	11.5	4.5	10.4	8.7	13.3	11.1	4.0	10.4	8.2	13.4	0.93 **	0.92 **
Magnesium (mg)	281.2	79.1	266.9	233.4	326.2	279.1	78.1	267.5	222.8	318.2	0.96 **	0.97 **
Manganese (mg)	3.8	1.2	3.6	3.0	4.3	3.7	1.2	3.6	2.9	4.3	0.93 **	0.92 **
Phosphorus (mg)	1070.0	272.5	1032.8	891.3	1251.2	1058.6	270.0	1021.1	864.6	1206.1	0.94 **	0.95 **
Potassium (mg)	2183.3	589.9	2183.3	1837.9	2603.0	2249.6	598.6	2166.7	1859.12	2572.0	0.95 **	0.95 **
Sodium (mg)	2393.3	582.3	2441.3	1842.6	2813.1	2381.0	545.3	2484.1	1829.9	2779.5	0.85 **	0.82 **
Zinc (mg)	8.6	2.2	8.2	7.0	9.9	8.5	2.2	8.1	6.9	9.6	0.95 **	0.93 **

Abbreviations: FFQ: Food Frequency Questionnaire; g: grams; ICCs: intraclass correlation coefficients; kcal: kilocalories; mg: micro-grams; SD: standard deviation. ** indicates significance at the 0.01 level.

**Table 4 nutrients-16-01132-t004:** Mean daily intakes of different food groups and intraclass correlation coefficients between the two FFQs administered to the study participants (*n* = 198).

Parameters	FFQ1					FFQ2					ICCs	
	Mean	SD	Median	25th	75th	Mean	SD	Median	25th	75th	Crude	Energy-Adjusted
Condiments (g)	16.4	5.9	21.3	10.7	21.3	18.6	5.5	21.3	10.7	21.3	0.81 **	0.81 **
Grains (g)	349.1	139.5	344.7	257.0	436.1	346.5	134.2	354.3	255.1	429.0	0.96 **	0.90 **
Fruits (g)	187.2	101.8	152.7	120.0	230.4	197.3	104.1	152.3	120.0	236.6	0.96 **	0.95 **
Vegetables (g)	245.7	122.3	240.0	170.0	310.3	266.2	120.9	236.4	165.5	315.0	0.94 **	0.93 **
Meat, Poultry, Processed Meat, and Organ Meat (g)	86.3	45.4	80.0	53.9	111.8	98.2	42.7	76.6	51.4	109.3	0.96 **	0.90 **
Fish and Seafoods (g)	45.9	31.6	43.7	23.1	58.0	50.6	33.4	41.6	19.8	57.9	0.89 **	0.88 **
Eggs (g)	37.7	21.3	38.8	25.0	50.0	35.4	22.2	40.0	25.0	50.0	0.94 **	0.94 **
Dairy and Dairy Products (g)	85.8	112.7	36.6	5.0	137.9	51.7	113.1	35.9	8.7	128.5	0.89 **	0.96 **
Beverages (mL)	57.6	144.5	19.4	0.0	61.9	89.8	142.9	27.0	0.0	85.6	0.95 **	0.95 **
Tea or Coffee (mL)	315.9	291.5	249.3	88.2	441.7	342.8	294.5	230.2	76.2	460.4	0.86 **	0.85 **
Soy and Soy Products (g)	78.4	85.6	50.9	21.0	92.4	52.9	83.5	44.8	23.2	96.6	0.93 **	0.93 **
Legumes (g)	15.5	19.1	7.2	3.6	21.6	10.1	15.1	8.2	3.6	18.9	0.76 **	0.75 **
Nuts and Seeds (g)	7.2	9.2	4.5	1.4	9.3	8.8	9.3	4.5	1.4	9.7	0.96 **	0.82 **
Sugary Snacks (g)	19.0	23.9	10.4	3.5	28.0	28.7	22.9	9.2	4.3	21.1	0.83 **	0.92 **
Water (mL)	1286.8	529.2	1295.4	967.8	1572.6	1282.9	526.4	1330.7	967.8	1572.6	0.92 **	0.92 **
Savory Snacks (g)	1.6	3.2	0.0	0.0	1.5	2.1	2.8	0.0	0.0	1.5	0.76 **	0.76 **
Dim Sum (g)	26.8	28.0	18.8	8.0	33.8	46.9	27.2	20.0	10.8	36.6	0.89 **	0.89 **
Oil (mL)	18.6	11.0	17.0	10.2	25.0	11.1	11.4	17.9	10.6	25.1	0.94 **	0.89 **
Sugars (g)	1.1	2.2	0.0	0.0	1.0	2.3	2.1	0.0	0.0	0.7	0.89 **	0.88 **
Wine (mL)	9.6	57.1	0.0	0.0	0.0	12.7	58.6	0.0	0.0	0.0	0.98 **	0.98 **

Abbreviations: FFQ: Food Frequency Questionnaire; g: grams; ICCs: intraclass correlation coefficients; mL: milliliters; SD: standard deviation. ** indicates significance at the 0.01 level.

**Table 5 nutrients-16-01132-t005:** Mean daily intakes of energy and selected nutrients and Pearson correlation coefficients between the FFQ1 and the average of three-day dietary records (*n* = 198).

Parameters	FFQ1					Three-Day DRs					Pearson Correlation Coefficients	
	Mean	SD	Median	25th	75th	Mean	SD	Median	25th	75th	Crude	Energy-Adjusted
Energy (kcal)	1541.2	334.0	1524.4	1321.6	1753.7	1669.0	375.7	1614.7	1410.3	1935.2	0.81 **	--
Energy from Total Fat (kcal)	509.6	147.9	488.0	397.0	589.9	572.4	173.1	554.5	443.3	682.6	0.75 **	--
Energy from Saturated Fat (kcal)	130.4	40.7	127.1	100.6	148.9	150.7	56.4	141.3	111.8	184.6	0.72 **	--
Energy from Trans Fat (kcal)	3.5	2.1	3.0	2.1	4.3	1.9	1.5	1.5	0.8	2.6	0.43 **	--
Protein (g)	73.8	17.9	72.2	61.0	83.7	79.1	20.6	77.4	63.9	91.7	0.76 **	0.69 **
Carbohydrates (g)	186.6	46.6	183.0	152.7	220.7	197.8	50.0	195.6	165.3	229.4	0.79 **	0.70 **
Total Dietary Fiber (g)	15.9	5.2	15.4	12.6	19.1	17.1	5.7	16.0	13.0	20.8	0.78 **	0.79 **
Total Sugar (g)	51.9	18.2	50.1	37.6	62.7	51.5	19.3	51.0	36.2	62.4	0.68 **	0.67 **
Total Fat (g)	56.9	16.6	54.3	44.3	65.8	63.8	19.3	61.7	49.5	76.1	0.75 **	0.71 **
Saturated Fat (g)	14.5	4.5	14.1	11.1	16.5	16.7	6.3	15.7	12.4	20.5	0.72 **	0.67 **
Trans Fat (g)	0.4	0.2	0.3	0.2	0.4	0.2	0.2	0.2	0.1	0.3	0.43 **	0.36 **
Cholesterol (mg)	290.2	93.1	286.1	218.6	353.1	347.9	126.0	337.8	254.0	418.3	0.66 **	0.67 **
Water (g)	2537.7	593.6	2488.9	2094.7	2894.9	2543.7	629.6	2500.2	2095.1	2869.8	0.73 **	0.73 **
Vitamin C (mg)	119.1	43.9	114.5	89.0	144.7	126.2	64.9	119.2	74.4	162.6	0.61 **	0.57 **
Calcium (mg)	685.9	224.9	631.5	547.0	803.5	636.1	265.8	580.9	437.6	767.2	0.69 **	0.67 **
Copper (mg)	1.0	0.3	0.9	0.7	1.1	1.2	0.4	1.1	0.9	1.4	0.64 **	0.62 **
Iron (mg)	11.5	4.5	10.4	8.7	13.3	10.5	4.1	9.5	7.7	12.3	0.65 **	0.62 **
Magnesium (mg)	281.2	79.1	266.9	233.4	326.2	276.2	87.5	254.7	215.8	327.7	0.76 **	0.77 **
Manganese (mg)	3.8	1.2	3.6	3.0	4.3	3.2	1.4	3.0	2.3	4.0	0.68 **	0.65 **
Phosphorus (mg)	1070.0	272.5	1032.8	891.3	1251.2	1058.1	281.1	1042.0	844.3	1232.0	0.72 **	0.65 **
Potassium (mg)	2183.3	589.9	2183.3	1837.9	2603.0	2429.0	704.2	2337.6	1981.4	2900.2	0.73 **	0.72 **
Sodium (mg)	2393.3	582.3	2441.3	1842.6	2813.1	3032.4	840.5	2922.3	2461.3	3573.3	0.44 **	0.37 **
Zinc (mg)	8.6	2.2	8.2	7.0	9.9	8.9	3.6	8.3	6.7	10.3	0.49 **	0.21 **

Abbreviations: DRs: dietary records; FFQ: Food Frequency Questionnaire; g: grams; kcal: kilocalories; mg: micro-grams; SD: standard deviation. ** indicates significance at the 0.01 level.

**Table 6 nutrients-16-01132-t006:** Mean daily intakes of different food groups and Pearson correlation coefficients between the FFQ1 and the average of three-day dietary records (*n* = 198).

Parameters	FFQ1					Three-Day DRs					Pearson Correlation Coefficients	
	Mean	SD	Median	25th	75th	Mean	SD	Median	25th	75th	Crude	Energy-Adjusted
Condiments (g)	16.4	5.9	21.3	10.7	21.3	18.6	9.9	16.1	12.4	21.3	0.19 **	0.18 *
Grains (g)	349.1	139.5	344.7	257.0	436.1	346.5	139.1	337.5	240.6	436.0	0.80 **	0.73 **
Fruits (g)	187.2	101.8	152.7	120.0	230.4	197.3	115.1	195.3	121.5	256.9	0.80 **	0.79 **
Vegetables (g)	245.7	122.3	240.0	170.0	310.3	266.2	149.1	242.8	161.6	347.7	0.75 **	0.75 **
Meat, Poultry, Processed Meat, and Organ Meat (g)	86.3	45.4	80.0	53.9	111.8	98.2	55.6	87.9	55.3	135.4	0.66 **	0.60 **
Fish and Seafoods (g)	45.9	31.6	43.7	23.1	58.0	50.6	43.6	42.4	15.6	76.7	0.53 **	0.52 **
Eggs (g)	37.7	21.3	38.8	25.0	50.0	35.4	26.0	33.3	16.7	50.0	0.74 **	0.74 **
Dairy and Dairy Products (g)	85.8	112.7	36.6	5.0	137.9	51.7	98.2	10.7	0.0	61.3	0.69 **	0.69 **
Beverages (mL)	57.6	144.5	19.4	0.0	61.9	89.8	136.5	8.1	0.0	161.7	0.52 **	0.51 **
Tea or Coffee (mL)	315.9	291.5	249.3	88.2	441.7	342.8	362.7	251.7	80.1	510.4	0.68 **	0.68 **
Soy and Soy Products (g)	78.4	85.6	50.9	21.0	92.4	52.9	84.7	12.4	0.0	80.0	0.66 **	0.66 **
Legumes (g)	15.5	19.1	7.2	3.6	21.6	10.1	21.8	0.0	0.0	10.0	0.27 **	0.24 **
Nuts and Seeds (g)	7.2	9.2	4.5	1.4	9.3	8.8	13.9	3.2	0.0	12.0	0.71 **	0.70 **
Sugary Snacks (g)	19.0	23.9	10.4	3.5	28.0	28.7	38.8	14.1	0.0	40.0	0.36 **	0.34 **
Water (mL)	1286.8	529.2	1295.4	967.8	1572.6	1282.9	508.8	1282.2	961.6	1602.7	0.84 **	0.84 **
Savory Snacks (g)	1.6	3.2	0.0	0.0	1.5	2.1	7.6	0.0	0.0	0.0	0.16 *	0.18 *
Dim Sum (g)	26.8	28.0	18.8	8.0	33.8	46.9	59.7	33.3	0.0	75.4	0.46 **	0.44 **
Oil (mL)	18.6	11.0	17.0	10.2	25.0	11.1	7.9	9.2	5.9	14.8	0.61 **	0.54 **
Sugars (g)	1.1	2.2	0.0	0.0	1.0	2.3	4.2	0.2	0.0	2.4	0.50 **	0.49 **
Wine (mL)	9.6	57.1	0.0	0.0	0.0	12.7	70.8	0.0	0.0	0.0	0.87 **	0.87 **

Abbreviations: DRs: dietary records; FFQ: Food Frequency Questionnaire; g: grams; mL: milliliters; SD: standard deviation. ** indicates significance at the 0.01 level. * indicates significance at the 0.05 level.

**Table 7 nutrients-16-01132-t007:** Cross-classification of energy and nutrient intake quartiles from FFQ1 and the average of three-day dietary records (*n* = 198).

Parameters	Crude ^a^			Adjusted for Energy ^a^		
	Same Quartile	Adjacent Quartile ^b^	Extreme Quartile ^c^	Same Quartile	Adjacent Quartile ^b^	Extreme Quartile ^c^
Energy (kcal)	61.6	33.3	5.1	--	--	--
Energy from Total Fat (kcal)	52.5	35.9	11.6	--	--	--
Energy from Saturated Fat (kcal)	46.0	40.9	13.1	--	--	--
Energy from Trans Fat (kcal)	44.4	35.4	20.2	--	--	--
Protein (g)	52.0	37.9	10.1	44.9	41.9	13.1
Carbohydrates (g)	62.6	32.3	5.1	37.4	32.3	30.3
Total Dietary Fiber (g)	49.5	41.4	9.1	53.0	39.4	7.6
Total Sugar (g)	43.4	43.9	12.6	42.9	41.9	15.2
Total Fat (g)	52.5	35.9	11.6	44.4	42.9	12.6
Saturated Fat (g)	46.0	40.9	13.1	51.5	35.9	12.6
Trans Fat (g)	44.4	35.4	20.2	34.8	46.0	19.2
Cholesterol (mg)	46.5	41.4	12.1	45.5	38.9	15.7
Water (g)	49.0	39.9	11.1	50.0	40.4	9.6
Vitamin C (mg)	39.4	40.4	20.2	33.3	48.5	18.2
Calcium (mg)	43.9	44.9	11.1	44.9	42.9	12.1
Copper (mg)	42.4	39.9	17.7	42.4	40.9	16.7
Iron (mg)	46.0	39.4	14.6	48.0	37.4	14.6
Magnesium (mg)	52.0	37.9	10.1	58.1	34.3	7.6
Manganese (mg)	51.5	37.4	11.1	50.0	38.4	11.6
Phosphorus (mg)	54.5	34.8	10.6	49.0	38.4	12.6
Potassium (mg)	51.0	36.4	12.6	54.0	32.8	13.1
Sodium (mg)	40.4	33.3	26.3	35.4	39.4	25.3
Zinc (mg)	47.5	36.4	16.2	41.4	32.8	25.8

^a^ Percentages from all categories may not round up to 100% exactly. ^b^ Same ± 1 quartile. ^c^ Same ± 2 quartiles. Abbreviations: g: grams; kcal: kilocalories; mg: micro-grams.

**Table 8 nutrients-16-01132-t008:** Cross-classification of food group intake quartiles from FFQ1 and the average of three-day dietary records (*n* = 198).

Parameters	Crude ^a^			Adjusted for Energy ^a^		
	Same Quartile	Adjacent Quartile ^b^	Extreme Quartile ^c^	Same Quartile	Adjacent Quartile ^b^	Extreme Quartile ^c^
Condiments (g)	36.4	37.4	26.3	27.8	39.4	32.8
Grains (g)	63.1	32.3	4.5	55.1	32.8	12.1
Fruits (g)	56.1	34.3	9.6	54.0	36.4	9.6
Vegetables (g)	53.5	36.4	10.1	53.0	35.4	11.6
Meat, Poultry, Processed Meat, and Organ Meat (g)	48.0	38.4	13.6	44.4	36.4	19.2
Fish and Seafoods (g)	38.9	37.4	23.8	41.4	38.4	20.3
Eggs (g)	59.6	20.7	19.7	40.9	46.5	12.6
Dairy and Dairy Products (g)	41.4	38.9	19.7	42.4	42.9	14.6
Beverages (mL)	36.9	25.8	37.3	42.4	30.8	26.8
Tea or Coffee (mL)	50.5	38.9	10.6	54.0	32.8	13.1
Soy and Soy Products (g)	38.9	31.8	29.3	34.8	40.9	24.2
Legumes (g)	29.8	29.8	40.4	35.9	36.4	27.8
Nuts and Seeds (g)	56.6	27.3	12.6	50.0	35.9	14.1
Sugary Snacks (g)	31.8	37.4	30.8	39.9	35.4	16.7
Water (mL)	65.5	27.0	7.4	68.2	24.2	7.6
Savory Snacks (g)	61.1	4.0	34.8	27.8	21.7	50.5
Dim Sum (g)	35.9	35.9	28.3	37.9	41.4	20.8
Oil (mL)	43.9	38.4	17.6	39.4	42.4	18.2
Sugars (g)	60.6	10.6	28.8	56.6	26.3	17.1
Wine (mL)	79.3	0	20.7	74.7	19.7	5.5

^a^ Percentages from all categories may not round up to 100% exactly. ^b^ Same ± 1 quartile. ^c^ Same ± 2 quartiles. Abbreviations: g: grams; ml: milliliters.

**Table 9 nutrients-16-01132-t009:** Results of the one-sample *t*-tests and linear regression analyses.

Parameters	Mean Difference	*p*-Value	Beta-Coefficient	*p*-Value
Nutrients				
Energy (kcal)	127.841	<0.001 *	0.195	0.006 *
Energy from Total Fat (kcal)	62.729	<0.001 *	0.231	0.001 *
Energy from Saturated Fat (kcal)	20.334	<0.001 *	0.429	<0.001 *
Energy from Trans Fat (kcal)	−1.690	<0.001 *	−0.370	<0.001 *
Protein (g)	5.204	<0.001 *	0.210	0.003
Carbohydrates (g)	11.153	<0.001 *	0.117	0.100
Total Dietary Fiber (g)	1.175	<0.001 *	0.143	0.045 *
Total Sugar (g)	−0.481	0.651	0.076	0.285
Total Fat (g)	6.842	<0.001 *	0.224	0.001 *
Saturated Fat (g)	2.259	<0.001 *	0.429	<0.001 *
Trans Fat (g)	−0.188	<0.001 *	−0.370	<0.001 *
Cholesterol (mg)	57.673	<0.001 *	0.380	<0.001 *
Water (g)	6.002	0.852	0.086	0.229
Vitamin C (mg)	7.078	0.056	0.449	<0.001 *
Calcium (mg)	−49.747	0.001 *	0.226	0.001 *
Copper (mg)	0.253	<0.001 *	0.330	<0.001 *
Iron (mg)	−0.992	<0.001 *	−0.126	0.076
Magnesium (mg)	−5.030	0.228	0.153	0.032 *
Manganese (mg)	−0.504	<0.001 *	0.139	0.051
Phosphorus (mg)	−11.948	0.419	0.045	0.529
Potassium (mg)	171.353	<0.001 *	0.252	<0.001 *
Sodium (mg)	639.140	<0.001 *	0.385	<0.001 *
Zinc (mg)	0.368	0.105	0.490	<0.001 *
Food Groups				
Condiments (g)	2.123	0.005 *	0.482	<0.001 *
Grains (g)	−2.582	0.684	−0.005	0.942
Fruits (g)	10.102	0.044 *	0.201	0.005 *
Vegetables (g)	20.539	0.004 *	0.288	<0.001 *
Meat, Poultry, Processed Meat, and Organ Meat (g)	11.886	<0.001 *	0.263	<0.001 *
Fish and Seafoods (g)	4.692	0.083	0.359	<0.001 *
Eggs (g)	−2.230	0.076	0.290	<0.001 *
Dairy and Dairy Products (g)	−31.135	<0.001 *	−0.188	0.008 *
Beverages (mL)	32.132	0.001 *	−0.067	0.349
Tea or Coffee (mL)	26.845	0.165	0.287	<0.001 *
Soy and Soy Products (g)	−25.521	<0.001 *	−0.015	0.838
Legumes (g)	−5.359	0.003 *	0.137	0.055
Nuts and Seeds (g)	1.642	0.020 *	0.517	<0.001 *
Sugary Snacks (g)	9.703	<0.001 *	0.476	<0.001 *
Water (mL)	−3.921	0.850	−0.073	0.304
Savory Snacks (g)	0.544	0.325	0.701	<0.001 *
Dim Sum (g)	20.077	<0.001 *	0.682	<0.001 *
Oil (mL)	−7.473	<0.001 *	−0.388	<0.001 *
Sugars (g)	1.259	<0.001 *	0.630	<0.001 *
Wine (mL)	3.083	0.220	−0.055	0.438

* indicates significant findings (*p*-value ≤ 0.05). Abbreviations: g: grams; kcal: kilocalories; mg: micro-grams; ml: milliliters.

## Data Availability

The data presented in this study are available on request from the corresponding authors. The data are not publicly available due to privacy or ethical restrictions.

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
