# Peer review of "Reproducibility and Relative Validity of a Short Food Frequency Questionnaire for Chinese Older Adults in Hong Kong"

_nutrients, 2024, doi:10.3390/nu16081132_

Round 1

Reviewer 1 Report

Comments and Suggestions for Authors

In this study the Authors aimed to assess the diet of Hong Kong older adults using the short FFQ and examine  its reproducibility and relative validity as a dietary assessment tool.

This paper is entirely constructed to demonstrate the reproducibility and relative validity of a short food frequency questionnaire, but the latter is not presented at all in the text or in the appendix.

It is therefore advisable to report all the questionnaires used to carry out the analysis.

The title and some parts of the paper are similar to those of another paper by the same author (Nutrients 2023, 15(12), 2668): it is advisable to modify them.

 Methods: since the conclusions report that the subjects were studied during the COVID-19 pandemic, this information must also be included in the Methods.

- line 87: “..200 older adults were included..”. Why then become 198?

- the deleted lines (lines 90-93) contain the references 16,17,18 which reported subsequently (line 104), appear after the references 19-21. The numbering needs to be adjusted

 Table 1: the average age of the entire population is identical to that of women (64 years), but men have an average age of 66 years: why?

Table 2: DRIs mean Dietary Reference Intakes? It is useful to include it in the legend

Tables 4,6,8,9: What does “Condiment” refer to, given that there is also oil?

 As the Authors rightly report, there is a great limitation due to the subjectivity of the three days and record dietary intakes

Conclusions: line 380: “trans fat” is repeated

Author Response

Reviewer 1

  1. In this study the Authors aimed to assess the diet of Hong Kong older adults using the short FFQ and examine its reproducibility and relative validity as a dietary assessment tool. This paper is entirely constructed to demonstrate the reproducibility and relative validity of a short food frequency questionnaire, but the latter is not presented at all in the text or in the appendix. It is therefore advisable to report all the questionnaires used to carry out the analysis.

Response: Thank you for your comment. We have presented the reproducibility and relative validity of our short FFQ as follows:

- Reproducibility: Intra-class correlation coefficients (Section 3.3.1.)

- Relative validity:
(1) Agreement between 2 methods by Bland-Altman analysis (Section 3.3.2.)
(2) Correlation analysis (Section 3.3.3.)
(3) Cross-classification of data reported by 2 methods (Section 3.3.4.)
(4) One-sample t-test and linear regression (Section 3.3.5.)

  1. The title and some parts of the paper are similar to those of another paper by the same author (Nutrients 2023, 15(12), 2668): it is advisable to modify them.

Response: Thank you for your comment. We have revised Section 4.2 paragraph 1 and 4.

  1. Methods: since the conclusions report that the subjects were studied during the COVID-19 pandemic, this information must also be included in the Methods.

Response: Thank you for your careful review. We have indicated it in Section 2.1 Study Design and Participants.

  1. line 87: “..200 older adults were included..”. Why then become 198?

Response: Thank you for your comment. Two participants had dropped out from this study due to personal issues and they did not complete the 2nd FFQ and three-day dietary records. We have also added explanation in paragraph 1 of Section 2.4 Statistical analysis.

  1. the deleted lines (lines 90-93) contain the references 16,17,18 which reported subsequently (line 104), appear after the references 19-21. The numbering needs to be adjusted

Response: Thank you for your careful review. We have updated the references accordingly.

  1. Table 1: the average age of the entire population is identical to that of women (64 years), but men have an average age of 66 years: why?

Response: Thank you for your comment. The average age of women should be “63.5” instead of “64”, as we had rounded it to the nearest integer before. We have updated the average age of women to 63.5 in Table 1. Also, the average age of the entire population was closer to that of women. It may because more women were included in this study.

  1. Table 2: DRIs mean Dietary Reference Intakes? It is useful to include it in the legend

Response: Thank you for your comment. We have added included the explanation in the legend of Table 2.

  1. Tables 4,6,8,9: What does “Condiment” refer to, given that there is also oil?

Response: Thank you for your comment. “Condiment” includes canned tomato sauce, canned and unsalted tomato paste, miso, dehydrated chicken broth, grounded cinnamon, spice blend, curry powder, grounded black pepper, grounded ginger, grounded turmeric, white pepper, ketchup, vinegar, sweet and sour sauce, barbecue sauce, teriyaki sauce, canned beef gravy, canned mushroom gravy, Worcestershire sauce, soy sauce, canned gravy with onions, hoisin sauce, oyster sauce, fish sauce, pesto, wasabi, spaghetti sauce, curry sauce, fermented and salted white soybean curd, fermented black bean, ginger puree with oil and salt, cheese or cream-based pasta sauce, commercial Asian satay sauce, fermented broad-bean paste, chili sauce, sesame paste and shrimp paste.

  1. As the Authors rightly report, there is a great limitation due to the subjectivity of the three days and record dietary intakes

Response: Thank you for your comment. Subjective bias is a common limitation of all dietary assessment methods as true intake of nutrients are difficult to measure. Biomarkers may allow objective measurement but not all nutrients have their own well-established biomarkers. Besides, the validation approach of this study aligns with previous studies which also used three-day dietary records as reference intake.

  1. Nielsen, D. E., Boucher, B. A., Da Costa, L. A., Jenkins, D. J. A., & El-Sohemy, A. (2023). Reproducibility and validity of the Toronto-modified Harvard food frequency questionnaire in a multi-ethnic sample of young adults. European journal of clinical nutrition, 77(2), 246–254. https://doi.org/10.1038/s41430-022-01209-z
  2. Syauqy, A., Afifah, D. N., Purwanti, R., Nissa, C., Fitranti, D. Y., & Chao, J. C. (2021). Reproducibility and Validity of a Food Frequency Questionnaire (FFQ) Developed for Middle-Aged and Older Adults in Semarang, Indonesia. Nutrients, 13(11), 4163. https://doi.org/10.3390/nu13114163
  3. Vioque, J., Gimenez-Monzo, D., Navarrete-Muñoz, E. M., Garcia-de-la-Hera, M., Gonzalez-Palacios, S., Rebagliato, M., Ballester, F., Murcia, M., Iñiguez, C., Granado, F., & INMA-Valencia Cohort Study (2016). Reproducibility and Validity of a Food Frequency Questionnaire Designed to Assess Diet in Children Aged 4-5 Years. PloS one, 11(11), e0167338. https://doi.org/10.1371/journal.pone.0167338

  1. Conclusions: line 380: “trans fat” is repeated

Response: Thank you for your comment. We would like to clarify that the phrases are “energy from trans fat” and “trans fat”, and therefore was not repeated.

Reviewer 2 Report

Comments and Suggestions for Authors

 For the validation, Authors included some methods but without a proper perspective. Authors should address the recommendations by Cade (https://www.ncbi.nlm.nih.gov/pubmed/12186666) and based on the recommendations they should present the results of the Bland-Altman method as the most important (as specified in recommendations: „The methods developed by Bland and Altman should be used to measure the agreement between food frequency questionnaires and other measures of dietary intake”). Taking this into account, such assessment should be made for all the assessed parameters of diet and should be present ed (either within main body of the study or supplementary materials). At the same time, the assessment conducted using the analysis of correlation has almost no value, so Authors should focus of the methods of the proven quality (even if they do not allow to indicate very high agreement between methods).

Author Response

Reviewer 2

  1. For the validation, Authors included some methods but without a proper perspective. Authors should address the recommendations by Cade (https://www.ncbi.nlm.nih.gov/pubmed/12186666) and based on the recommendations they should present the results of the Bland-Altman method as the most important (as specified in recommendations: „The methods developed by Bland and Altman should be used to measure the agreement between food frequency questionnaires and other measures of dietary intake”). Taking this into account, such assessment should be made for all the assessed parameters of diet and should be present ed (either within main body of the study or supplementary materials).

Response: Thank you for your suggestion. We have moved the Bland-Altman results from Section 3.3.5. to 3.3.2. to emphasize its importance. Besides, we have summarized the Bland-Altman analysis results of all nutrients parameters in Table S1 and visualized them as Figure S1-S16.

  1. At the same time, the assessment conducted using the analysis of correlation has almost no value, so Authors should focus of the methods of the proven quality (even if they do not allow to indicate very high agreement between methods).

Response: Thank you for your comment. We have followed the study approach of previous validation papers. Correlation analysis is one of the common methods for validation studies, some examples are provided:

  1. Nielsen, D. E., Boucher, B. A., Da Costa, L. A., Jenkins, D. J. A., & El-Sohemy, A. (2023). Reproducibility and validity of the Toronto-modified Harvard food frequency questionnaire in a multi-ethnic sample of young adults. European journal of clinical nutrition, 77(2), 246–254. https://doi.org/10.1038/s41430-022-01209-z
  2. Syauqy, A., Afifah, D. N., Purwanti, R., Nissa, C., Fitranti, D. Y., & Chao, J. C. (2021). Reproducibility and Validity of a Food Frequency Questionnaire (FFQ) Developed for Middle-Aged and Older Adults in Semarang, Indonesia. Nutrients, 13(11), 4163. https://doi.org/10.3390/nu13114163
  3. Vioque, J., Gimenez-Monzo, D., Navarrete-Muñoz, E. M., Garcia-de-la-Hera, M., Gonzalez-Palacios, S., Rebagliato, M., Ballester, F., Murcia, M., Iñiguez, C., Granado, F., & INMA-Valencia Cohort Study (2016). Reproducibility and Validity of a Food Frequency Questionnaire Designed to Assess Diet in Children Aged 4-5 Years. PloS one, 11(11), e0167338. https://doi.org/10.1371/journal.pone.0167338